# Quality of life in people living with HIV-associated neurocognitive disorder: A scoping review study

Kate Alford[1]*, Stephanie Daley[2], Sube Banerjee[3], Jaime H. Vera[1,4,5]

**1** Department of Global Health and Infection, Brighton and Sussex Medical School, University of Sussex, Falmer, East Sussex, United Kingdom, **2** Centre for Dementia Studies, University of Sussex, Falmer, East Sussex, United Kingdom, **3** Faculty of Health, University of Plymouth, Plymouth, United Kingdom, **4** Department of Medicine, Brighton and Sussex Medical School, University of Sussex, Falmer, East Sussex, United Kingdom, **5** Brighton and Sussex University Hospitals NHS Trust, Brighton, East Sussex, United Kingdom

* K.Alford2@bsms.ac.uk

**Data Availability Statement:** All data are contained within the manuscript and supporting information files.

**Funding:** The study was funded as part of a PhD fellowship for KA by Brighton and Sussex Medical

## Abstract

Quality of life (QoL) is recognized as an essential end point in the disease management of chronic conditions such as HIV with calls to include good QoL as a 'fourth 90' in the 90-90-90 testing and treatment targets introduced by World Health Organization in 2016. Cognitive impairments impact a broad spectrum of experiences and are a common issue effecting people living with HIV (PLWH). Despite this, few studies have examined QoL in PLWH who also have a cognitive disorder. This study aimed to synthesize and describe what is known about QoL in those living with HIV-associated neurocognitive disorders (HAND). A scoping review of peer-reviewed literature was conducted to identify how QoL has been investigated and measured in PLWH with HAND, and how PLWH with HAND report and describe their QoL. We searched PsychInfo, Medline, Scopus, and Web of Science along with hand-searching reference lists from relevant studies found. Included studies were those published in English after 1ˢᵗ January 2003 which included PLWH with cognitive impairment not due to other pre-existing conditions. Fifteen articles met criteria for inclusion. Two studies measured QoL as a primary aim, with others including QoL assessment as part of a broader battery of outcomes. The MOS-HIV and SF-36 were the most commonly used measures of overall QoL, with findings generally suggestive of poorer overall QoL in PLWH with HAND, compared to PLWH without cognitive impairment. Studies which examined dimensions of QoL focused exclusively on functionality, level of independence, and psychological QoL domains. There is a considerable dearth of research examining QoL in PLWH with HAND. The initiatives which advocate for healthy aging and improved QoL in PLWH must be extended to include and understand the experiences those also living with cognitive impairment. Research is needed to understand the broad experiential impacts of living with these two complex, chronic conditions, to ensure interventions are meaningful to patients and potential benefits are not missed.

School, The Centre for Dementia Studies, and Sussex Partnership NHS Foundation Trust. The funders had no role in the study design, data collection and analysis, decision to publish, or preparation of the manuscript.

**Competing interests:** J H V reports honoraria and research grants in trials sponsored by Merck, Janssen Cilag, Piramal and Gilead Sciences. SB reports grants and personal fees from Abbvie, personal fees and non-financial support from Lilly, personal fees from Eleusis, personal fees from Daval International, personal fees from Boehringer-Ingelheim, personal fees from Axovant, personal fees from Lundbeck, personal feesfrom Nutricia, outside the submitted work; he has been employed by the Department of Health for England. This does not alter our adherence to PLOS ONE policies on sharing data and materials.

# Introduction

Worldwide approximately 38 million people live with human immunodeficiency virus (HIV) with 1.7 million new cases seen each year [1]. While life expectancy has greatly increased due to effective combination antiretroviral therapy (cART), cognitive disorders remain a significant issue for people living with HIV (PLWH) and those who care for them. HIV-associated dementia, which was seen in up to 50% of PLWH in the pre-cART era, is now rare, however, a mild to moderate profile of cognitive impairment impacting quality of life (QoL), medication adherence, and employment is widely reported [2–5].

HIV-associated Neurocognitive Disorder (HAND) is the term coined to refer to clinically significant declines in multiple domains of neuropsychological functioning that are not exclusively attributable to factors other than HIV infection. For research purposes, the gold-standard Frascati Criteria [6] is commonly used to identify HAND across three thresholds of impairment (asymptomatic Neurocognitive Impairment (ANI); mild neurocognitive Disorder (MND); and HIV-associated Dementia (HAD)). Frascati Criteria–based HAND involves testing across broad cognitive domains, with HAND identified when two or more domains are one or more standard deviations below normative scores and for MND and HAD when daily functioning is impacted. Notably, for a clinical diagnosis of HAND (based on Diagnostic and Statistical Manual of Mental Disorders (5th ed; DSM-5) [7], impairment must be reported both objectively (e.g., decline in standard neuropsychological testing) and subjectively (e.g., complaints), whereas evidence of subjective impairment is not required for Frascati Criteria-based HAND identification. Additionally, there must be no evidence of a pre-existing cause for impairment. Prevalence rates vary depending on definitions of cognitive impairment used, with studies reporting cognitive impairment in as many as 52% of PLWH [8], which may or may not be related to HIV. Those with more stringent definitions estimate that between 14–28% of PLWH experience these issues [9]. Prevalence of cognitive impairment, however, is expected to rise as the HIV-population continues to age, whereby the impacts of aging and associated morbidities compound the cognitive vulnerabilities already seen in PLWH.

With 73% of PLWH in Europe estimated to be over 50 years of age by 2030, an emerging consensus that we need to measure broad patient-rated outcomes such as quality of life (QoL) along with discrete areas of function like cognition and mental health, has emerged. Examination of QoL allows healthcare providers a more thorough understanding of the factors impacting PLWH and facilitates the provision of optimum person-centred care [10]. QoL is commonly conceived as dynamic, subjective and multidimensional, with dimensions often including physical, psychological, social, and spiritual factors [11] relating to 'an individuals' perception of their position in life in the context of the culture and value systems in which they live and in relation to their goals, expectations, standards and concerns' [12]. Assessment of QoL has been widely embraced in generic HIV clinical care, with the World Health Organization (WHO) endorsing QoL as an important and prominent patient-reported outcome in clinical care and intervention efficacy, along with HIV advocacy groups calling for its inclusion as an additional 90 in the 90-90-90 testing and treatment target introduced by the World Health Organization in 2016 [13]. However, PLWH with HAND have been recognised as an underserved population in healthcare [14]. To date much of the research into HAND has focused on etiology, diagnosis and treatment (e.g. improved management of clinical variables contributing to impairments) with little consideration for the experiential consequences of having two chronic, stigmatizing, and potentially debilitating conditions.

There is value in examining the literature to identify what is currently known about the relation between QoL in PLWH with HAND to provide a better overall picture of the experiences of these individuals. Mapping what is known about the QoL of those living with HIV

with HAND, this review will identify the gaps in knowledge allowing the formulation of new research requirements. To the best of our knowledge, a synthesis or summary of the evidence pertaining to QoL in PLWH with HAND has not been previously conducted. Therefore, a mapping or scoping (i.e. summarising the range of evidence to report breadth and depth) of the research conducted will contribute to a better understanding of the situation. Here we describe the extent, range, and nature of the body of knowledge on QoL in PLWH with HAND for the purpose of providing a systematic, synthesised summary of the evidence. Examining the research on QoL in those PLWH with HAND will help provide a basis for mitigating negative outcomes, such as health risk and burdens for both those with HAND and their families/carers, along with interventions to advance wellbeing and QoL. With this in mind the research questions are:

1. How has QoL been investigated and measured with regard to PLWH with HAND?,

2. How do PLWH with HAND report and describe their QoL?

## Methods

A scoping review was selected based on preliminary searches showing that studies examining QoL in PLWH with HAND were lacking, with the majority of studies found either not examining QoL (or any dimension associated with QoL) or not look at those PLWH with HAND. Munn et al. [15] states the utility of scoping reviews, as opposed to systematic reviews, when examining limited, disparate, or emerging evidence, and when broad as opposed to specific questions are posed. These indications allow for questions regarding types of available evidence, how the research has been conducted, key factors or characteristics of a topic, and the identification of gaps in knowledge, and as such, was considered most appropriate for this research topic. A scoping review was conducted with the assistance of an experienced librarian (A.F) using systematic search methods. Our objectives were to document the available evidence describing the methods used to measure and capture QoL experiences, along with describing how PLWH with HAND report their QoL. Our choice of review method was further informed following initial searches in Google Scholar and the database Scopus. This preliminary search showed that studies examining QoL in PLWH with HAND were lacking, with the majority of studies found either did not examine QoL (or any dimension associated with QoL) or did not look at those PLWH with HAND. The framework employed to structure and inform this review was developed by Arksey and O'Malley [16], with further guidance from later publications, including the Preferred Reporting Items for Systematic Reviews and Meta-Analyses Extension for Scoping Reviews (PRISMA-ScR) (S1 Table) [17, 18]. It consists of six stages: identifying the research question; searching for relevant studies; deciding on studies for inclusion; charting the data; collating, summarizing, and reporting study findings; and consulting stakeholders [16]. Study limitations resulted in no stakeholder consultation for this review, however, all other steps were followed. So as to allow a broad map of the literature there was an agreement to include articles which describe a multi-dimensional evaluation of QoL and to extract and report associations related to QoL that were statistically significant (i.e. 95% Confidence Level).

### Identifying relevant studies

We conducted a broad search of peer-reviewed literature. Searches were conducted of four electronic databases: Scopus, Web of Science, Medline and PsychInfo, to identify peer-reviewed literature, with the last searches performed on 25th September 2020. Date limit was

set at studies published from 1st January 2003 onwards, due to this being the time with which access to cART became more widely available. Searches involved analysis of text words contained in the title and abstract, and of the index terms used to describe the article. A second search using all identified keywords and index terms was then conducted across all databases. Our search strategy was designed to be broad and sensitive enough to ensure we capture all relevant studies (Table 1) and included specifications of the *context* (PLWH), *participant* (HAND) and *concept* (quality of life). The *context* terms consisted of HIV, AIDS, HIV/AIDS, Human immunodeficiency virus. The *participants* term included HAND, HIV-associated neurocognitive impairment, cognitive impairment, neurocognitive impairment. The term *concept* consisted of quality of life, health-related quality of life, well-being, life satisfaction and functioning. In addition, reference lists of relevant studies and reviews found were also hand-searched to ensure there was comprehensive coverage of the research topic.

Table 1 presents the search strategy that was used for Scopus, which was adapted in minor ways for other databases.

## Study selection

**Inclusion and exclusion criteria.** Quantitative and qualitative peer-reviewed, original research papers were included, reviews, commentaries and editorials were excluded. Interventional studies were excluded, as it is possible that the mechanism with which the intervention impact QoL complicates the interpretation of the outcome and will affect the clarity of inferences drawn. Given the aim included summarising knowledge status, grey literature was not included as the findings had not been published in peer-reviewed journals. Only articles presented in English were considered for inclusion.

**Context.** Studies must state that participants either were on Highly active antiretroviral therapy (HAART) or cART or had access to HAART or cART, to this end only studies dated 1st January 2003 onwards were considered for review, as this is when the availability of HAART/cART became widespread. Prior to the advent of effective ARVs, the impacts HIV/AIDS and antiretroviral therapies had on QoL was hugely significant [19], this review seeks to consolidate QoL for those with HAND in the era of effective treatment (i.e. HAART or cART).

**Participants.** Studies had to include populations over 18 years. Studies considered for inclusion must have demonstrated that the cognitive impairment seen in the sample are primarily due to HIV-associated factors (i.e. HAND) e.g. the primary reason for impairments is not a mental illness, Alzheimer's dementia, or cardiovascular disease. For this to be evident,

**Table 1. Search strategy terms.**

| Search terms | |
|---|---|
| Context | Concept |
| #1 HIV (263,618) | #10 "quality of life" (425,651) |
| #2 AIDS (155,378) | #11 QoL (40,042) |
| #3 HIV/AIDS (76,220) | #12 "health-related quality of life" (44,122) |
| #4 "Human immunodeficiency*" (242,879) | #13 H?QoL (16,556) |
| #5 (#1 or #2 or #3 or #4) (377,902) | #14 "well-being" (135,714) |
| Participants | #15 "life satisfaction" (16,941) |
| #6 HAND (830,550) | #16 function* (6,240,621) |
| #7 "HIV-associated neuro*" (1,664) | #17 (#10 or #11 or #12 or #13 or #14 or #15 or #16) (147,602) |
| #8 "Cognitive impairment" (71,264) | |
| #9 Neurocognitive* (22,041) | **Combined search:** |
| #10 (#6 or #7 or #8 or #9) (918,754) | #5 and #10 and #17 (217) |

studies needed to demonstrate that the population selected for study has a diagnosis of HAND, which is based on the current 'gold standard' Frascati criteria [6]. Therefore, studies must have detailed evidence that the population sampled had i) acquired impairment in cognitive functioning, involving at least two domains, documented by performance of at least 1 standard deviation below the mean for age-education-appropriate norms on standardized neuropsychological tests, and ii) impairment decidedly due to primarily HIV-related factors, with no evidence of another pre-existing cause likely to be primarily responsible for the cognitive impairments seen.

**Concept.** The key concept is QoL, including its multidimensional components. With this in mind studies examining the components of QoL will be considered for review. For the purpose of this review components included are physical health, psychological state, functionality, level of independence, social relations, environment, and spirituality/religion/personal beliefs [12]. While, functionality is not considered to comprise the QoL concept by the WHO [12], other authors argue it be a relevant and informative proxy or dimension of QoL [20, 21] (19), which has been validated in HIV populations [22]. With this in mind, studies examining functionality (e.g. activities of daily living; ADL) will be considered for review.

**Search results.** The search strategy in the four different databases resulted in a total of 1005 articles being identified. These records were exported into EndNoteX8. Two articles were identified through hand-searching relevant articles reference list. The removal of duplicate records resulted in 543 records left for title and abstracts screening. The screening was performed by one author (KA), under the close guidance of the second author (JV) who was consulted where inclusion was unclear. KA and JV finally agreed to 92 records as relevant studies for full-text screening. KA and JV then independently screened full-text studies to assess eligibility for inclusion in the review. KA screened the reference lists of the included studies. Of the 92 records considered for inclusion, there was disagreement on 2 (2.2%), in which cases discussion meetings were held until an agreement was reached. Of the 92 screened full-text articles, 78 were excluded. The majority of articles were not considered to be examining HAND i.e. cognitive impairment was not distinguished from other potential causes, or participants were not assessed using a neuropsychological test battery and impairments were subjective. A number of studies did not examine QoL in any depth (i.e. a single-item QoL question was employed and no other QoL domains were examined or had ineligible study populations) or it could not be confidently concluded participants had access to effective ARV therapy. A total of 15 articles were included in the review (Fig 1).

### Charting the data

Information relating to QoL in PLWH with HAND were examined by two authors (KA and JV) who utilized processes from qualitative content analysis [23]. This required three phases of analysis: preparation, organization, and reporting, which was conducted using a structured data tool. KA extracted the study characteristics, which was then reviewed by JV, and any conflicts were discussed until resolution was reached. Given this was a scoping review, study quality (e.g. possible bias, sample sizes) was not evaluated [16]. Tables 2 and 3 detail the systematization and categorization of study characteristics and results included in the review, based on the research questions.

### Collating, summarising, and reporting the results

Study characteristics and main findings of the 15 studies included in this review are presented in Table 2. Studies are ordered by year of publication, with those published in the same year grouped in order of author surname.

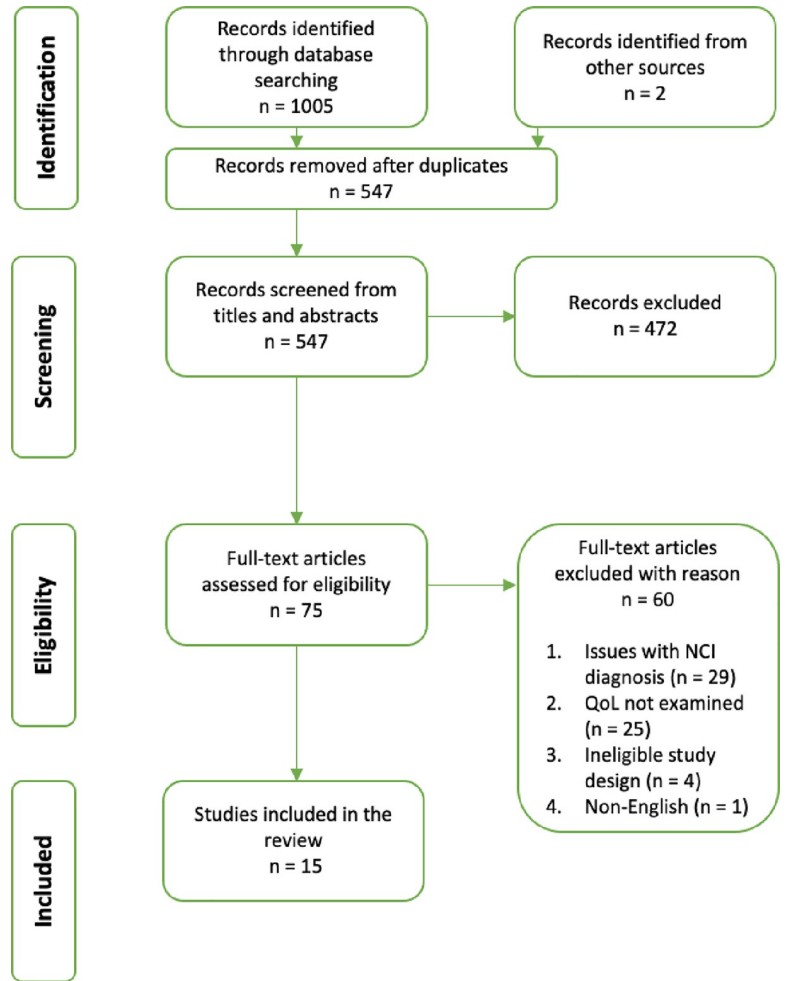

**Fig 1. PRISMA flowchart.**

## Results

The results of this review are reported in line with the review questions. Of the 1005 peer-reviewed abstracts, 75 were advanced for full-text review which yielded 15 articles that met inclusion criteria (Fig 1). The results section first describes the study characteristics (Table 2) and details how QoL has been assessed in this population, we then describe how PLWH with HAND report their QoL.

Eight studies were conducted in the United States [24, 31, 32, 35, 38, 40, 42, 47], three studies were conducted in Canada [45, 49, 51], two were conducted in Italy [27, 30], one study conducted in Brazil [50], and one in the United Kingdom [43]. The primary aim of the majority of studies was to examine functional deficits associated with HAND, which was assessed via 'real-world' functional assessments [24, 31, 32, 47] or through patient-reported outcome measures (PROMs) of functioning [24, 31, 32, 35, 40, 47], with broad QoL assessment or the assessment of a second QoL component an additional secondary aim in some studies [24, 32, 38]. Four studies examined the impact of HAND on health-related QoL (HRQoL) [27, 30, 38, 45], two studies assessed psychological state in those with HAND [50, 51], one study examined the impact of positive health behaviours on HAND [42], one study examined associations between different PROMs and HAND [43], and one study explored how HAND impacted well-being

**Table 2. Study characteristics and main findings.**

| Reference, year, and country | Aims and Methods (inc. instruments used) | Comparison group | Sample size and population | Test method for associations between HAND and QoL | QoL domains examined | QoL perspectives from results section |
|---|---|---|---|---|---|---|
| 1. Mindt *et al.* [24] 2003 USA | • To investigate the cultural relevance of a pre-existing functional battery for a Spanish population<br>• To explore the relationship between cognitive impairment (CI) and the ability to perform ADL<br>• Cross-sectional methodology<br>• Functional assessment of ADLs (capacity-based ADL assessment)<br>• Katz Adjustment Scale; KAS-P (Functional QoL) [25]<br>Patient's Own Assessment of Functioning Inventory; PAOFI (Functional measure) [26] | Comparisons between PLWH with and without HAND | N = 16 (N HAND = 8) Age/gender not given | Chi-square tests to compare functioning across groups | • Functionality<br>• Level of Independence | • Those with HAND were significantly more functionally impaired compared to those without HAND<br>• PLWH with HAND reported significantly lower QoL expectation compared to controls<br>• Performances on functional assessments and CI status were related to indicators of real-world functioning inc. employment and QoL |
| 2. Tozzi *et al.* [27] 2003 Italy | • To examine the relationship of HAND and health-related QoL (HRQoL)<br>• Cross-sectional study<br>• MOS-HIV (HRQoL measure) [28]<br>• Karnofsky Performance Scale; KPS (functional measure inc. health status) [29] | Comparisons between PLWH with and without HAND | N = 111 (N HAND = 37) 31 female and 80 male (HAND: 6 female and 31 male) Mean age: 37 years (HAND: 38 years) | • Chi-square test to examine comparisons between groups<br>• Linear regression to assess associations between neuropsychological domains and QoL measures | • Pain<br>• Physical functioning<br>• Role functioning<br>• Social functioning<br>• Mental health<br>• Energy<br>• Health distress<br>• Cognitive functioning<br>• General health perceptions<br>• Overall QoL<br>• Functionality | • Those with HAND report significantly poorer QoL in all sub-domains of MOS-HIV including a physical health summary score and mental health summary score<br>• Those with HAND report significantly worse KPS scores<br>• Those with more severe CI had greatest probability of lower QoL |
| 3. Tozzi *et al.* [30] 2004 Italy | • To determine the association of CI with HRQoL in PLWH on HAART<br>• Cross-sectional study<br>• MOS-HIV (HRQoL measure) [28]<br>• Karnofsky Performance Status; KPS (functional measure inc. health status) [29] | Comparisons between PLWH with and without HAND | N = 70 (N HAND = 23) 21 female and 49 male (HAND: 4 female and 19 male) Mean age: 36 years (HAND: 38 years) | • Chi-square test to examine group comparisons<br>• Multivariable logistic regression model to identify variables independently related to HAND | • Pain<br>• Physical functioning<br>• Role functioning<br>Social functioning<br>• Mental health<br>• Energy<br>• Health distress<br>• Cognitive functioning<br>• General health perceptions<br>• Overall QoL<br>• Functionality | • CI is significantly associated with lower HRQoL in all sub-domains of the MOS-HIV and Karnofsky performance status<br>• Lower physical health summary scores and mental health summary scores (MOS-HIV) was associated with presence of HAND |

*(Continued)*

**Table 2.** (Continued)

| Reference, year, and country | Aims and Methods (inc. instruments used) | Comparison group | Sample size and population | Test method for associations between HAND and QoL | QoL domains examined | QoL perspectives from results section |
|---|---|---|---|---|---|---|
| 4. Heaton *et al.* [31] 2004 USA | • To evaluate the functional and 'real world' impacts of HAND in PLWH • Cross-sectional study • Functional 'real-life' capacity tests (i.e. cooking, shopping, financial management, medication management) • Lawson & Brody Instrumental Activities of Daily Living Scale— Modified (IADL) (functional assessment) [31] | Comparisons between PLWH with and without HAND | N = 267 (N HAND = 99) 224 male and 43 female (HAND: 79 male and 20 female) Mean age: 39 years (HAND: 39 years) | • Chi-square tests of association, t-tests and ANOVAs to examine comparisons between groups • Multivariate analyses | • Level of independence • Functionality | • Presence of HAND and impairment on functional tests significantly associated with subjective experiences of cognitive difficulties, unemployment and increased dependence in ADL • Impairment on IADL test and depression were the only unique predictors of 'real-world' functioning |
| 5. Gandhi *et al* [32] 2010 USA | • To compare the results of new performance-based functional assessments for the study of HAND in PLWH and examine the relationship between functional assessments and severity levels of HAND • Cross-sectional study • Columbia Medication Management test (CMMT) [31] • San Diego Finances test (SDFT) [33] • Karnofsky Performance Status (KSP) (functional measure inc. health status) [29] • MOS-HIV sub-domains: physical QoL, role QoL [28] • Beck Depression Inventory -II (BDI-II) [34] | Comparison across HAND diagnosis severity i.e. ANI, MND and HAD and those without HAND | N = 114 (N HAND = 98) 76 male and 38 female (HAND: 64 male and 34 female) Mean age: 47 years (HAND: 47 years) | • Chi-square tests of association to examine group differences • Logistic regression to assess ability of the functional performance measures to differentiate between strata on rating scales | • Level of independence • Physical QoL • Functionality • Psychological state | • Individuals classified as having HIV-dementia were more likely to have symptoms of depression • Better performance on CMMT, SDFT or KSP associated with decreased odds of HAND, while poorer scores on MOS-HIV sub-domains associated with increased odd of HAND • Significant differences between those with and those without HAND on Role and Physical QoL (MOS-HIV), with role and physical QoL reported to be lower in those with HAND |
| 6. Cattie *et al.* [35] 2012 USA | • To examine to everyday functional implications of planning dysfunction in PLWH with HAND • Case controlled study Tower of London^DX (Neuropsychological planning test) [36] • Profile of Mood States (POMS) (mental health symptomatology assessment) [37] • Lawson & Brody IADL Scale–modified (functional assessment) [31] | Comparisons between PLWH with and without HAND and HIV- controls | N = 244 (N HAND = 53) 46 female and 198 males (HAND: 9 female and 44 male) Mean age: 46 years (HAND: 45 years) | Planned regression analysis and logistic regression | • Level of independence • Functionality • Psychological state | • Participants with HAND show impairments in areas of planning which were significant independent predictors of self-reported decline in ADL inc. employment status. • Those with HAND report significantly worse mental health (POMS) than those PLWH without HAND and HIV- controls |

(*Continued*)

**Table 2.** (Continued)

| Reference, year, and country | Aims and Methods (inc. instruments used) | Comparison group | Sample size and population | Test method for associations between HAND and QoL | QoL domains examined | QoL perspectives from results section |
|---|---|---|---|---|---|---|
| 7. Morgan *et al.* [38] 2012 USA | • To determine whether HIV infection and aging act synergistically to disrupt everyday functioning<br>• Case-controlled study<br>• 36-item Short-form Health Survey (SF-36) (HRQoL assessment) [39]<br>• Lawson & Brody IADL Scale–modified (functional assessment) [31]<br>• Karnofsky Performance Scale; KPS (functional measure inc. health status) [29] | Presence of HAND examined across different factors across different groups (HIV-Young, HIV- Old, HIV + young, HIV+ Old) | N = 109 (N HAND = 31) 81 male and 30 female (HAND gender not given) Mean age = 43 years (HAND mean age not given) | Multiple linear regression models | • Level of independence<br>• Functionality | • Presence of HAND in PLWH was a significant predictor of ADL declines<br>• Presence of HAND did not predict poorer reported HRQoL. Older PLWH showed disproportionately worse emotional functioning, with current depressive disorder sole risk factor for lower HRQoL. |
| 8. Iudicello *et al.* [40] 2013 USA | • To examine the magnitude, cognitive correlates, and everyday functioning impact of risky decision-making impairments in PLWH with HAND<br>• Case-controlled study<br>• Iowa Gambling Task (IGT) (Neuropsychological measure of risk taking) [41]<br>• Beck Depression Inventory-II; BDI-II (assessment of depressive symptomatology) [34] Lawson and Brody IADL Scale -Modified [31] | Comparisons between PLWH with and without HAND and HIV- controls | N = 146 (N HAND = 68) 132 male and 14 female (HAND: 62 male and 6 female) Mean age = 43 years (HAND: 44) | • T-tests to explore significance across study group by IGT functional domain<br>• Logistic regression analysis to explore potential predictors of adverse functional outcomes | • Level of independence<br>• Functionality<br>• Psychological state | • Findings suggest riskier decision-making in HAND but risker decisions making did not predict functional outcomes<br>• Significant predictors of ADL dependence in PLWH was presence of HAND<br>• Unemployment status was associated with HAND diagnosis<br>• Significantly higher instance of depressive symptoms in those with HAND compared to those without and HIV- controls |
| 9. Fazeli *et al.* [42] 2014 USA | • To examine if proxy measures of engagement in a number of 'lifestyle' factors (ALFs) (i.e. physical exercise, social activity and employment) are independently associated with better neurocognitive functioning in PLWH<br>• Case-controlled study<br>• Patient reported outcome measure (PROM) of physical exercise, social activity and employment<br>• Karnofsky Performance Status; KPS (functional measure inc. health status) [29] | Comparisons between PLWH with and without HAND | N = 139 (N HAND not given) Mean age = 49 years (HAND not given) 111 male and 28 female (HAND not given) | Multiple logistic regression model predicting presence of HAND using ALF classification | • Level of independence<br>• Physical health<br>• Social relationships<br>• Functionality | • Significant independent main effects of ALF classification, i.e. a higher number of ALFs is associated with a lower presence of HAND. Suggesting those with HAND engage in less physical exercise, social activity and employment. |

(*Continued*)

**Table 2.** (Continued)

| Reference, year, and country | Aims and Methods (inc. instruments used) | Comparison group | Sample size and population | Test method for associations between HAND and QoL | QoL domains examined | QoL perspectives from results section |
|---|---|---|---|---|---|---|
| 10. Underwood et al. [43] 2016 UK | • To examine the predictive validity of different PROMs in PLWH with HAND and how this may vary based of HAND diagnostic method • Case-controlled study • HAND defined according to the Frascati Criteria, Global Deficit Score (GDS) and Multivariate Normative Comparison (MNC) • Lawson & Brody IADL Scale–modified (functional assessment) [31] • Short-Form Health Survey; SF-36 (HRQoL measure) [39] • Patient Health Questionnaire; PHQ-9 (assessment of depressive symptomatology) [44] | Comparisons between PLWH without HAND and demographically matched HIV- controls | N = 387 (N HAND (Frascati classified) = 87) Median age = 57 (HAND not given 313 male and 67 female (HAND not given) | Wilcoxon rank-sum tests to examine differences in physical and mental health between those with and without HAND | • Level of independence • Functionality • Psychological state • Physical health | • Patients both with and without HAND report high rates of symptomology (i.e. functional difficulties) making associations challenging • Only PHS and MHS sub-scores were reported (from SF-36). These were lower for those with HAND but only significantly so for MHS. |
| 11. Tymchuk et al. [45] 2017 Canada | • To assess the prevalence and severity of depressive symptoms in a representative cohort of HIV/AIDS patients in Canada and investigate the risk factors for depression inc. neuropsychological performance, HAND, and demographic and clinical variable • Cross-sectional • EVGGPF QoL Tool (single item measure of HRQoL) [46] • Patient's Health Questionnaire; PHQ-9 (assessment of depressive symptomatology) [44] | Comparisons between PLWH in different depression severity groups (mild, moderate, severe) | N = 268 235 male and 33 female Mean age = 47 years (No descriptive given for those diagnosed with HAND) | ANOVA and Chi-Squared for group comparisons. Multinominal and binary regression to test all significant variables predicting depressive symptomatology | • Psychological state • HRQoL | • Highest proportion of those with HAND in moderate/severe depression groups. However, multivariate analyses showed a lack of association between depressive symptomatology and HAND |
| 12. Woods et al. [47] 2017 USA | • To examine the extent to which HAND in PLWH interferes with internet-based household ADLs • Case controlled study • Online functional skills tests (shopping, banking) • Lawson & Brody IADL Scale–modified (functional assessment) [31] • UCSD Performance-based skills assessment [48] • Karnofsky Performance Status; KPS (functional measure inc. health status) [29] Patient's Own Assessment of Function Inventory; PAOFI (functional assessment) [26] | Comparisons between PLWH with and without HAND and HIV- controls | N = 134 (N HAND = 43) 18 female and 116 male (HAND: 6 female and 37 male) Mean age: 45 years (HAND: 45 years) | ANOVA post-hoc comparisons for continuous functional scores and chi-square tests for dichotomous outcomes | • Functionality • Level of independence | • PLWH with HAND were significantly more likely to fail the online shopping task and performed worse on the online banking tests, compared to PLWH without HAND and HIV- controls • In the HIV+ sample, lower scores across both internet-based tasks were uniquely associated with poorer performance-based functional capacity and self-reported declines in shopping and financial management in daily life. |

*(Continued)*

**Table 2.** (Continued)

| Reference, year, and country | Aims and Methods (inc. instruments used) | Comparison group | Sample size and population | Test method for associations between HAND and QoL | QoL domains examined | QoL perspectives from results section |
|---|---|---|---|---|---|---|
| 13. Terspsra *et al.* [49] 2018 Canada | • To explore how PLWH with HAND view, manage and obtain support for their cognitive difficulties with a focus on how cognitive difficulties manifested, progressed and impacted well-being<br>• Qualitative research study<br>• Semi-structured interview methodology | No comparison | N = 25 (all PLWH with HAND)<br>20 males and 5 females<br>Median age = 51 years | Thematic analysis of transcripts | • Physical health<br>• Psychological state<br>• Social relationships<br>• Environmental influences<br>• Level of independence<br>• Functionality | • Participants reported cognitive difficulties which interfered with activities at home, school and work<br>• Difficulties fulfilling ADLs<br>• Participants recollected missing or forgetting meals, medications and medical appointments<br>• Participants described feeling 'stupid' and reported sadness with regard to how 'they used to be'<br>• The inherent stress, worry, and negative thoughts regarding their CIs worsened abilities to remember and caused emotional upset<br>• External stressors impacted their cognitive status<br>• Participants reported subjective links to low mood and cognitive abilities<br>• Participants collectively described experiences of embarrassment, frustration and worry regarding potential declines and further loss of independence<br>• Reports of stigma (felt stigma, self-stigma and enacted stigma)<br>• Importance of having social support<br>• Many participants felt their CI prevented them from forming and maintaining social relationships |
| 14. Gascon *et al.* [50] 2018 Brazil | • Sought to examine prevalence, demographic, clinical and neuropsychological characteristics and QoL outcomes of PLWH in Sao Paulo, Brazil<br>• Cross-sectional study<br>• Beck Depression Inventory-II; BDI-II (assessment of depressive symptomatology) [34] | Comparisons between PLWH with and without HAND | N = 412 (N HAND = 303)<br>281 male and 131 female (HAND: 204 male and 99 female)<br>Mean age = 45 years<br>(HAND = 46 years) | Chi-square tests to examine group differences. | • Psychological state<br>• Functionality | • Higher BDI scores were significantly associated with HAND<br>• Those with HAND significantly more likely to be unemployed |

*(Continued)*

**Table 2.** (Continued)

| Reference, year, and country | Aims and Methods (inc. instruments used) | Comparison group | Sample size and population | Test method for associations between HAND and QoL | QoL domains examined | QoL perspectives from results section |
|---|---|---|---|---|---|---|
| 15. Tu *et al.* [51] 2020 Canada | • To investigate the prevalence and associated variables in PLWH with and without HAND<br>• Case-controlled study<br>• EVGGPF QoL Tool (single item measure of HRQoL) [46]<br>• Patient's Health Questionnaire; PHQ-9 (assessment of depressive symptomatology) [44] | Comparisons between: PLWH with no CI, PLWH with HAND and PLWH with CI due to other disorders (not primary HIV causation) | N = 370 (N HAND = 78) 323 male and 47 female (HAND = 70 male and 8 female) Mean age = 47 years (mean age HAND = 47 years) | Univariate and multivariate (logistic regression and Random Forest analyses) to assess group differences | • HRQoL<br>• Psychological State | • Significant differences found between reported HRQoL and PHQ-9 scores: PLWH with CI due to other disorders reported worse HRQoL and higher levels of depressive symptoms than those without CI and those with HAND. |

broadly [49]. In six of the studies patients were recruited from HIV outpatient clinics [32, 35, 38, 45, 50, 51], five studies recruited patients from ongoing research cohorts [24, 31, 40, 43, 47], two studies recruited from community settings [42, 49], and two studies recruited from a neurology clinic [27, 30]. Fourteen studies employed quantitative methodologies: seven studies were case-control studies [35, 38, 40, 42, 43, 47, 51] and seven has a cross-sectional design [27, 30–32, 45, 50]. One qualitative study was included, which employed interpretative qualitative methods [49].

## Population

One study was conducted with PLWH with HAND exclusively [49], this is likely due to the qualitative methodology used. Seven studies [24, 27, 30, 42, 45, 50] included a single comparator group; PLWH without HAND, five studies [31, 35, 38, 40, 43, 47] included two comparator groups; PLWH without HAND and HIV-negative controls, and one study [32] examined differences across HAND severity (e.g. No cognitive impairment, ANI, MND, and HAD), and one study [38] examined differences between older and younger PLWH and HIV negative controls, with HAND status being a factor examined among other variables.

Participants totaled 2286, of these 1052 had diagnosed HAND. Of those with HAND 595 participants were male and 192 were female. Gender distinctions for those with HAND were not available for 4 studies [24, 38, 42, 43], this was due to the primary aim not being outcome across HAND categorization (HAND vs. no HAND) [24, 38, 42] or multiple methods of HAND diagnosis being examined with gender not distinguished between methods [43]. None of the studies report whether neuropsychological testing considered gender distinctions for normative scores or whether it was considered in diagnosing HAND, however this would normally only be examined in association analyses which seek to examine gender distinctions in HAND. The average age of HAND participants was approximately 44 years. An exact calculation based on all studies included was not possible as mean age was not reported for those with HAND [42, 43, 45] or median age was given [43]. The majority of studies did not provide ethnicity data for those with HAND [24, 27, 30, 38, 42, 43, 45, 50, 51]. In studies which did Caucasian/white participants comprised 45.8%, Black/African/African-American/Caribbean participants representing 41.2%, and Hispanic participants comprising 8.2% of the sample. All but one study [49] gave information pertaining to the ARV status of participants; four studies stated that all participants were on stable cART [32, 38] or HAART [27, 30], and across the remaining studies 86.8% of participants were on cART or HAART. One study was included in

**Table 3. QoL measures and dimensions.**

| Instrument | QoL domains examined | Study inclusion |
|---|---|---|
| Medical Outcomes Survey–MOS-HIV (35-item) [52] | Physical functioning | [27, 30] |
| | Role functioning | [32]—Role functioning items and the MOS physical function subscale only) |
| | Pain | |
| | Social functioning | |
| | Emotional well-being | |
| | Energy/fatigue | |
| | Cognitive functioning | |
| | General health | |
| | Health distress | |
| | Overall QoL | |
| SF-36 (6-items) [39] | Physical functioning | [38, 43] |
| | Role Functioning (physical and emotional) | |
| | Bodily pain | |
| | General health | |
| | Vitality | |
| | Social functioning | |
| | Mental health | |
| EVGGFP (Single-item HRQoL measure) [46] | HRQoL | [45, 51] |
| Katz Adjustment Scale–patient KAS-P (3-item) [25] | Physical functioning | [24] |
| | Global QoL expectation | |
| | Health satisfaction | |
| Patient's Own Assessment of Functioning Inventory PAOFI (41-items) [26] | Functioning (domains examined: memory, language and communication, dexterity, sensory-perceptual, higher level cognitive abilities) | [24, 47] |
| | Level of Independence | |
| Karnofsky Performance Scale Index (11-point rating scale ranging from normal functioning to dead) [29] | Physical functioning | [27, 30, 32, 38, 42, 47] |
| | Level of independence | |
| Lawson & Brody Instrumental Activities of Daily Living–Modified (8-items) [31] | Physical functioning | [31, 35, 38, 40, 43, 47] |
| | Level of independence | |
| | Health status | |
| | (Functional ability across 8 common daily, 'complex' tasks (Telephone use, Shopping, Food preparation, Housekeeping, Laundry, Transportation, Medication management and Financial management) | |
| ALF Questionnaire (9-items) [32] | Physical functioning | [32] |
| | Mental functioning | |
| | Social functioning | |
| PHQ-9 (9-item) [44] | Mental health | [43, 45, 51] |
| Beck Depression Inventory BDI-II (21-Item) [53] | Mental health | [32, 40, 50] |
| Profile of Mood States POMS (65-items) [37] | Mental wellbeing | [35] |

this review [49], despite not providing ARV status information (likely due to its qualitative methodology and that it was not deemed relevant for the objectives of the research) however, contact was made with the study authors who confirmed all participants had access to cART.

## QoL measures

Measures used to assess QoL are listed in Table 3. A relatively small number of multidimensional QoL instruments were used. Only four studies used an instrument with the capacity to

examine QoL broadly [27, 30, 38, 43]. The MOS-HIV [52], modified and validated for use in PLWH, was utilized by two studies [27, 30]; it measures ten dimensions of QoL and allows the user to calculate a physical (PHS) and mental health summary score (MHS) and an overall QoL score. Likewise, the newer Short-Form Survey 36 (SF-36) [39], was employed by two studies [27, 30, 38, 43], and can be described both as a generic health instrument and a health-related QoL tool. It covers eight domains, and allows the administer to calculate a PHS, MHS, and overall QoL score. Two studies; [45, 51] used the EVGGFP [46], a single-item measure of HRQoL, which asks patients to rate their HRQoL today, and yields moderate validity for assessing HRQoL in PLWH [46].

A number of studies, in addition to the broader QoL instruments aforementioned [30, 43, 47], used PROMs to examine manifest functioning. Lawson & Brody Instrumental Activities of Daily Living Scale—modified [31, 54] was used by six studies [31, 35, 38, 40, 43, 47] and the Patient's Own Assessment of Functioning Inventory (POAFI) [26] was used by three studies [24, 40, 47]. These measures broadly assess day-to-day functional abilities (e.g. medication management, cooking) and patients' perceptions of a wide array of functional abilities (e.g. memory, language and communication). Four studies [30, 38, 42, 47] included the Karnofsky Performance Scale (KPS) [29], which provides a broad indication of functional ability and insight into overall health status. One study [24] used the Katz Adjustment Scale (KAS-P) [25] which examines functional ability, along with evaluating health perceptions and health satis-faction. A purpose-build assessment measure of 'Active Lifestyle Factors' was developed by one study [42] to examine physical health, social activity and mental activity (via proxies of physical exercise, social connectiveness and employment).

Measures of psychological state were employed by eight studies, all of which focused on depression and anxiety symptomatology using the Patient's Health Questionnaire (PHQ-9) [43–45, 51], Beck Depression Inventory II (BDI- II) [31, 32, 50], and the Patient's Own Mood State (POMS) [35].

## How do PLWH with HAND report their QoL?

**QoL overall.** Studies found overall QoL was reduced in those with HAND compared to controls [27, 30, 38, 43, 51], however, this was significant in only two studies [27, 30]. Global QoL expectation was significantly lower in those with HAND compared to those without a cognitive impairment [24]. Experientially PLWH with HAND described day-to-day fluctuations, and 'good days and bad days' in moods and abilities, which yielded prominent downstream effects on their perceptions of QoL.

**Physical health.** PLWH with HAND reported summative worse physical health summary QoL scores [27, 30, 43]. In addition, pain and energy QoL sub-domains were found to be significantly worse in PLWH with HAND [27, 30]. These findings were corroborated by qualitative reported of poor sleep mediating cognitive disfunction, missed or forgotten meals, problematic medication adherence, and missed medication appointments which likely impact physical health and wellbeing [49].

**Psychological.** Overall studies which examined psychological QoL [27, 30, 32, 35, 40, 43, 45, 50, 51] indicate that PLWH with HAND experience lower rates of psychological QoL. Three studies found those with HAND reported lower MHS and poorer mental health QoL [27, 30, 43]. Studies which additionally employed a mental health measure [32, 35, 40, 43, 45, 50, 51] found higher instances of depressive and anxious symptoms in those with HAND compared to controls [35, 40, 43, 50] and a positive association between depressive symptomology and severity of cognitive impairment [32]. Qualitatively, participants reported the broad impacts of HAND on their psychological QoL, including its impact on self-esteem, self-

concept, identity, mood (including stress, worries, depressive/anxious symptoms), anxieties regarding the future, and self and felt stigma.

**Level of independence.** Studies reported negative relationships between level of independence and presence of HAND on QoL [38]. HAND impacts complex or instrumental ADLs such as cooking, shopping, medication and financial management, using transport etc. [24, 31, 35, 38, 40, 47]. In addition to predicting [38], and being associated with [24, 47] declines in ADLs, one study reported 67% of those with a HAND diagnosis to be considered functionally impaired [31] and those with HAND were significantly more likely to be classed as 'dependent' [40, 47]. Experientially, participants described how difficulties with ADLs affected their QoL as it lessened their ability to be independent.

**Social relationships.** Social QoL was examined in two studies [27, 30] as part of a broad QoL measure, with both reporting social QoL to be significantly lower in PLWH with HAND. No other studies examined this QoL domain quantitatively. Qualitatively, participants described difficulties forming and maintaining friendships due to their cognitive impairment, which in turn, impacted their self-esteem [49].

**Environment.** The impact of HAND on environmental QoL in PLWH was not examined quantitively by any study, but was touched upon by participants in the qualitative study [49]. Participants recalled how stressful life events appeared to worsen their cognitive symptoms and some felt their cognitive difficulties put them at risk in certain situations and impacted their ability to work.

**Spirituality/Religion/Personal beliefs.** We found no reports pertaining to QoL in relation to spirituality, religion or personal beliefs.

## Discussion

In chronic conditions 'adding life to years' is as essential as 'adding years to life' [55]. QoL has increasingly been recognised as a valuable patient-reported outcome which is now more routinely measured in HIV clinical care and within memory services. QoL research in these populations dictates the importance of specificity when applying QoL measurement [56, 57]: evidence from PLWH and from those with cognitive impairment finds unique dimensions influence QoL in these populations [22, 58–60]. Failure to capture and assess the relevant domains of QoL results in interventions which may lack meaning to patients along with potentially helpful interventions being undervalued, or the potential deleterious impacts of interventions being missed. This review provides an important first step in understanding how QoL is impacted for those PLWH with HAND by synthesizing what is currently known and identifying what is currently missing.

It is striking that across the literature we found only two studies focused on QoL exclusively as their main research outcome [27, 30]. These two studies found that QoL, as evaluated by the MOS-HIV, was significant worse for PLWH with HAND than PLWH without HAND, both overall and across a variety of QoL domains (all p < 0.04). An important consideration with regard to these findings is that these studies were conducted during the very early stages of effective ARV availability, with both studies reporting average time on effective ART as below 30 months. Therefore, while it can confidently be asserted that HAND impacts QoL, this finding is possibly relatively time specific, and must be applied with caution to today's patients–who, in the vast majority of cases have been on effective ARV therapy for decades, of which the cumulative impact of considerably better health and the downstream impact on QoL, must be considerable.

Another finding across the majority of studies, is the impact HAND appears to have on psychological QoL. The quantitative findings regarding psychological QoL appear to suggest there

is a significant impact of HAND on psychological QoL: three studies [27, 30, 43] employing standardised QoL instruments found a significant negative impact of HAND in PLWH *vs.* PLWH without HAND [27, 30] and HIV- controls [43]. However, it must be noted that the only recent study [43] (conducted in 2017 *vs.* 2003/4 [27, 30]), which examined this using a standardised QoL instrument found a difference between groups when HAND was categorised using the Multivariate Normative Method [61], and not when using Frascati Criteria diagnostics. The authors note that this may be explained by the considerably mild and asymptomatic participant sample used in the study, and thus finding should be generalised carefully [43]. Mental health measures conducted by studies overall found higher instances of depressive and anxious symptoms in those with HAND compared to controls [35, 40, 43, 50]. Given the robust and well evidenced relationship between psychiatric symptoms and QoL [62] this provides more tentative evidence for an important effect of HAND on psychological QoL. This is an important area for future research: it is well documented that PLWH report significantly higher levels of depression than HIV-negative controls [63], any additive impact of HAND on psychological QoL is important to understand so that psychological interventions in those with HAND can address the psychological factors most relevant to improving psychological QoL in this group.

A further key finding was the overwhelming focus on functional assessment seen in the studies included. All but one study [43] (the authors note the nature of HAND in their study was very mild, which likely effected ability to detect group differences in outcomes) found significant differences in functional ability and its impact on independent living. Areas of functioning examined by studies included instrumental ADL (Lawson & Brody IADL Scale [31, 54] or Lawson & Brody IADL Scale-modified [31, 54]), broad physical functioning ability and level of independence (Karnofsky Performance Scale [29]) and self-reported functional abilities with regard to memory, language and communication, dexterity, sensory-perceptual, and higher-level cognitive abilities (POAFI [26]). Studies show functional disability has been demonstrated consistently both within and across measures reflecting a wide array of functional skills, strongly suggesting functional deficits impacting ability to live independently are seen commonly in PLWH with HAND. Importantly, however, the functional impairments seen in those with HAND appear to be relatively mild, with impacts primarily seen on complex ADLs. Indeed, most studies did not provide contextual information pertaining the functional meaning of scores, however relatively few participants were reported to have diagnosis of HAD which is diagnosed *based* on severe functional impairments. Future research is needed to understand the impact poor functioning and lessening independence has on QoL broadly. This is particularly important given the relatedly mild form of HAND now most commonly seen, and given the push to enhance person-centred HIV care, which demands a holistic understanding of difficulties so that overall health is assessed along with the social, psychological, and environmental aspects of HIV and poor cognition that can lead to negative outcomes [64].

Significantly, there is the considerable lack of research in this field. Other than the impact of HAND on psychological QoL, functional ability, and impact on level of independence no other domains of QoL have been investigated in any depth, if at all. Of note, is the lack of research into the social effects of living with HAND. Indeed, there is a the wealth of literature describing increased levels of loneliness in PLWH (particularly older PLWH) [65] and in those with cognitive difficulties [66], and the downstream impact this has broad domains of QoL. The importance of social connectedness for good QoL is evidenced to be pivotal in other clinical populations [67, 68] and interventions addressing social factors such as loneliness have shown good efficacy in PLWH. It would certainly be of value to better understand the mechanisms which may be driving the poorer reported social QoL reported in this group [27, 30], so that interventions which build this skill and provide social support can be developed. Similarly,

very little information was found on how one's environment impacts QoL and no studies reported on the spiritual, religious, or existential dimensions of QOL purported to be important to overall QoL by the WHO [69]. More research is needed to identify what may serve as protective or mitigating factors on QoL, as this will be important to understand if we are seeking to improve it. It is also of interest that no studies have examined the intersectional impact, of having both HIV and HAND, two potentially stigmatised and long-term health conditions, on QoL and their joint effects on health and wellbeing. Qualitative research is particularly valuable here, and while the qualitative study included in this review [49] has shone light on some of these issues (e.g. participants reported struggling to maintain and form new friendships, how instances of enacted stigma have impacted their confidence), more qualitative studies looking at QoL directly would certainly be a beneficial first step in explicating the meaningful constituents of QoL, the relationships between different variables, and the protective factors on QoL in this group of patients.

The studies reviewed in this scoping review were predominately conducted in the USA, however studies from a range of other countries were seen. Notably, no studies were conducted in Africa, where the burden of HIV is highest, and samples were heavily weighted toward male participants, despite no there being on gender disparity in PLWH with HAND. Ethnicity of participants with HAND was infrequently reported, particularly in studies where examination of HAND was not a primary aim. The impact of demographic variables on QoL in this population is an important area of research missing from this field: in HIV clinical care, demographic variables provide important clues to guide intervention development and implementation. Interestingly, a recent meta-analysis examined the sociodemographic impacts on QoL in PLWH: finding low socioeconomic status, stigma, less education, and younger age to negatively influence QoL, and increased social support to positively influence QoL [70]. This study emphasises the importance of understanding sociodemographic influences on QoL in PLWH with HAND particularly given the intersectional and compounding issues facing these individuals. Studies were relatively evenly distributed across the time period, suggesting there has not been increased interest in the area in recent times, this is in contrast to the increasing attention QoL has been receiving in other chronic conditions, including HIV broadly [71]. One qualitative study met our inclusion criteria, all other studies examined QoL and/or domains of QoL quantitatively. Most studies did not explore QoL in PLWH with HAND as a primary aim, with comparisons between groups on QoL measures often part of a broad battery of outcome variables assessed with relative brevity. The majority of studies used established and generic instruments when examining QoL and its associated dimensions. Studies which assessed QoL broadly, used either the MOS-HIV [52], the SF-36 [39] or the EEVGGP [46]. The MOS-HIV and SF-36 can be compared with relative consistency, as both measures are based on the same concept and theoretical perspective of QoL [72]. However, neither of the two studies [38, 43] which used the SF-36, report sub-domain scores and only one study [43] reports MHS and PHS scores, making detailed conclusions challenging. Furthermore, unlike the MOS-HIV, the SF-36 does not assess cognitive functioning, which is likely an important area to assess this group of patients. We included studies which measured dimensions of QoL– most examined were function and psychological state. The use of measures for both dimensions was fairly homogenous: with the majority of studies employing either Lawson & Brody IADL Scale [54] or Lawson & Brody IADL Scale -Modified [31], Karnofsky Performance Status and Patient's Own Assessment of Function Inventory [29], as measures of function (albeit domains of functioning varied greatly across measures (see Table 3), and the Beck Depression Inventory-II [34] and the Patient's Health Questionnaire-9 [44], as measures of psychological state. Given the homogeneity across studies with regard to comparison groups used (i.e. HIV-and/or PLWH with HAND) comparisons between studies can broadly be drawn.

The main strength of this research is that it is the first review exploring and synthesising what is known about the impact of HAND on QoL in PLWH: taken cumulatively the findings appear to indicate a significant impact on QoL–which needs to be considered both with regard to patient clinical care and increased research attention. A second strength of this scoping review is the systematic way in which it was performed. The search was conducted with comprehensive search terms, which identified a large number of studies across four leading databases. Although the search was not exhaustive, the selected databases and the searches performed were advised by experienced academic librarians in order to generate the widest possible search for texts related to the population, concept, and context. Furthermore, we applied stringent eligibility criteria to the assessment of cognitive impairment in studies included, ensuring that the research 'gold-standard' Frascati criteria [6] was used to assess HAND. Limitations of this review include the paucity of studies from sub-Saharan Africa, where the burden of HIV is highest, along with a lack of examination of the sociodemographic factors impacting QoL in this population. Furthermore, important consideration must be paid to the context in which this research appears, whereby screening and diagnosing HIV-associated neurocognitive impairments remains relatively limited in clinical settings [2, 73], with no consensus on screening tools and concerns regarding the importance of routine testing [74, 75], given the majority of instances are asymptomatic and services extremely limited [76].

## Conclusions

QoL in PLWH with HAND is a significantly under reported field in HIV clinical care. Few studies have examined QoL in this group of patients and in those that have the impact on function and subsequent level of independence of PLWH with HAND on QoL have been reported most frequently. Studies appear to suggest PLWH with HAND do report lower QoL overall, however more research is needed to confidently assert this. PLWH with HAND appear to experience greater levels of functional impairments, albeit mildly, which likely impact their QoL to varying extents. Psychological QoL appears to be lower in those with HAND, however whether this manifests into clinical illness is undetermined. This may represent an important target for interventions as improvements to mental health is likely lead to improved cognition, and improved QoL. More research is needed to understand the degree to which QoL is impacted by HAND, the dimensions of QoL which may be effected, and whether standard conceptualisations (or, indeed, HIV or dementia specific models) of QoL capture QoL experiences for this group, so that variables of interest can be identified and meaningful interventions developed to improve QoL in this population. QoL outcomes have received increased focus over the last two decades, and recent calls to include QoL in The WHO's HIV/AIDS 90-90-90 target is justified based on multiple sources of evidence finding significant impacts of living with HIV on QoL [77–79]. While there is more need for research exploring QoL in those living with HAND, the emerging evidence detailed in this review along with evidence from those studies reporting poorer QoL in PLWH broadly, shows that a paradigm shift is required when thinking about those living with HIV. Clinical care needs to reflect the realities of patients' lives and multidimensional support is required to ensure treatment effects are maximized and reflective of factors most important to patients.

## Supporting information

**S1 Table. Preferred reporting items for systematic reviews and meta-analyses extension for Scoping Reviews (PRISMA-ScR) checklist.**
(DOCX)

## Acknowledgments

We thank the staff at Brighton and Sussex Medical School Library for assistance with the systematic search.

## Author Contributions

**Conceptualization:** Kate Alford, Stephanie Daley, Sube Banerjee, Jaime H. Vera.

**Data curation:** Kate Alford, Jaime H. Vera.

**Formal analysis:** Kate Alford, Jaime H. Vera.

**Investigation:** Kate Alford, Jaime H. Vera.

**Methodology:** Kate Alford, Stephanie Daley, Jaime H. Vera.

**Project administration:** Kate Alford.

**Supervision:** Stephanie Daley, Sube Banerjee, Jaime H. Vera.

**Writing – original draft:** Kate Alford.

**Writing – review & editing:** Kate Alford, Stephanie Daley, Sube Banerjee, Jaime H. Vera.

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
