## [Decision Letter · Decision Letter 0]

13 Apr 2021

PONE-D-21-03610

Quality of life in people living with HIV-associated neurocognitive disorder: A scoping review

PLOS ONE

Dear Dr. Alford,

Thank you for submitting your manuscript to PLOS ONE. After careful consideration, we feel that it has merit but does not fully meet PLOS ONE’s publication criteria as it currently stands. Therefore, we invite you to submit a revised version of the manuscript that addresses the points raised during the review process.

We look forward to receiving your revised manuscript.

Kind regards,

Simone Reppermund, PhD

Academic Editor

PLOS ONE

Journal Requirements:

2.  Thank you for stating the following in the Disclosures section of your manuscript:

"J H V reports honoraria and research grants in trials sponsored by Merck, Janssen Cilag, Piramal and

 Gilead Sciences. SB reports grants and personal fees from Abbvie, personal fees and non-financial

support from Lilly, personal fees from Eleusis, personal fees from Daval International, personal fees

from Boehringer-Ingelheim, personal fees from Axovant, personal fees from Lundbeck, personal fees

from Nutricia, outside the submitted work; he has been employed by the Department of Health for

England. These funders had no role in the design of the study; in the collection, analyses, or

interpretation of data; in the writing of the manuscript, or in the decision to publish the results."

Please remove any funding-related text from the manuscript and let us know how you would like to update your Funding Statement.

Reviewers' comments:

Reviewer's Responses to Questions

**Comments to the Author**

1. Is the manuscript technically sound, and do the data support the conclusions?

Reviewer #1: Yes

Reviewer #2: Yes

2. Has the statistical analysis been performed appropriately and rigorously? 

Reviewer #1: Yes

Reviewer #2: N/A

3. Have the authors made all data underlying the findings in their manuscript fully available?

Reviewer #1: Yes

Reviewer #2: Yes

4. Is the manuscript presented in an intelligible fashion and written in standard English?

Reviewer #1: Yes

Reviewer #2: Yes

5. Review Comments to the Author

Reviewer #1: Review for manuscript titled: Quality of life in people living with HIV-associated neurocognitive disorder: A scoping review.

Journal: Plos One

General comments. This review examines a very important, and often neglected, aspect of living with HIV-associated cognitive impairment. The authors use the phrases “PLWH with HAND” and “PLWH and HAND” interchangeably. Either or are fine, but can the authors please be consistent? There are some minor grammatical errors throughout the manuscript.

Introduction. The literature review is well structured and the authors covers a substantive amount of relevant literature in the field of HIV-associated cognitive disorders and how this may affect quality of life. The extensive literature review paves the way for a strong rationale.

Methodology. The methods followed are clear and replicable. The sub-heading “search strategy” on line 174 should be changed to “search results” or “search strategy outcomes” since the actual results of the search strategy are discussed in this section. The search strategy itself is described earlier in the methods section. The authors use the PRISMA diagram to visually represent their results and also provide the PRISMA checklist for scoping review, but have chosen a framework by Arksey and O’Malley (2005) to guide their methodology. Please can the authors elaborate on this choice and explain why they deem it appropriate to apply this framework in conjunction to the PRISMA statement. Please can the authors include in the manuscript, a statement on what makes a scoping review different from a systematic review when the ultimate goal of both is to describe the depth and breadth of the state of science in a particular area/topic and how this review fits into that definition. In the PRISMA diagram, please check for a typo in the block “records after duplicates removed n=460”. This number should be 547 as these are the records AFTER duplicates were removed. Alternatively change the text in the block to read “duplicate records removed n =460”. The image quality of the PRISMA diagram is not very good, please can the authors provide a 600dpi image or use the word template for PRISMA diagrams provided on the PRISMA statement website.

Results. Table 2 of the results provides a very good overview of the studies included and their findings. The authors report that 4 of the included studies did not include gender distinctions. Given that some neuropsychological tests take gender into consideration when doing the scaled scoring it would be important to check which neuropsychological tests these studies used to classify HAND. Regardless of the study aims, standard neuropsychological test scoring according to the test manuals needs to be observed, so it may be that these studies did not report the gender distinctions as opposed to it being a methodological matter. For the Canadian study that was included regardless of the treatment status of the participants, it is confirmed that the data reported in that study was collected in the accessible ART era even though the publication date was 2018?

Discussion. The discussion is well written and provides a good dialogue and argument for including quality of life into routine care for PLWH both with and without cognitive impairment. What the discission is missing is the closing of the circle for the case to include quality of life into the 90-90-90 WHO targets. The authors identify the main strength of this review as being the systematic way in which it was conducted. While I agree that this is a strength of this review, I feel that this secondary to the very important dialogue this review has paved way for. The results, and the argument created in the discussion make a very strong case indeed for having quality of life added into the 90-90-90 targets and this should be emphasized more in the discussion. I do not agree that the stringent criteria applied to the assessment of cognitive impairment is a limitation of this study. This should be added to the strengths of the study because these criteria ensure that a proper gold standard method was used to asses HIV-associated cognitive disorders in the included studies. A possible limitation of this study is the paucity of studies from sub-Saharan Africa. The impact of socioeconomic status on quality of life has not been acknowledged, and while other reviews have been excluded from the search strategy, the authors should acknowledge a recent review and meta-analysis published in 2020. Although this review does not specifically assess quality of life in HIV+ persons with cognitive impairment, important assertions are made regarding social and other determinant of quality of life in PLWH.

Ghiasvand, Hesam, Peter Higgs, Mehdi Noroozi, Gholamreza Ghaedamini Harouni, Morteza Hemmat, Elahe Ahounbar, Javad Haroni, Seyran Naghdi, Ali Nazeri Astaneh, and Bahram Armoon. "Social and demographical determinants of quality of life in people who live with HIV/AIDS infection: evidence from a meta-analysis." Biodemography and social biology 65, no. 1 (2020): 57-72.

Reviewer #2: This is a well written and important study that provides an essential review of the current literature and data about how having HIV neurocognitive impairment (NCI) affects quality of life (QoL). This is an important and thorough review that will inform the field. There are, however, some issues that need addressing. First, HIV Associated Neurocognitive Disorder (HAND) is not a clinical diagnosis. It is used for research purposes. The authors should make this important distinction. ICD-10 or DSM-V clinical diagnoses for HIV-associated neurocognitive problems require different criteria than the Frascati criteria. Second, the authors may wish to consider discussing that screening and diagnosing HIV associated NCIs remain very limited in many clinical settings [see for example 1-6] as do services for HAND [see for example 7].

1. Barber TJ, Bradshaw D, Hughes D, Leonidou L, Margetts A, Ratcliffe D, Thornton S, Pozniak A, Asboe D, Mandalia S, Boffito M, Davies N, Gazzard B, Catalan J. Screening for HIV-related neurocognitive impairment in clinical practice: Challenges and opportunities. AIDS Care. 2013:1-9.

2. Haddow LJ, Accoroni A, Cartledge JD, Manji H, Benn P, Gilson RJC. Routine detection and management of neurocognitive impairment in HIV-positive patients in a UK centre. International Journal of STD & AIDS. April 30, 2013 2013.

3. Morley D, McNamara P, Kennelly S, McMahon G, Bergin C. Limitations to the identification of HIV-associated neurocognitive disorders in clinical practice. HIV Medicine. 2013.

4. The Mind Exchange Working Group. Assessment, Diagnosis, and Treatment of HIV-Associated Neurocognitive Disorder: A Consensus Report of the Mind Exchange Program. Clinical Infectious Diseases. April 1, 2013 2013;56(7):1004-1017.

5. Kim S, Ades M, Pinho V, Cournos F, McKinnon K. Patterns of HIV and mental health service integration in New York State. AIDS care. 2014 Aug 3;26(8):1027-31.

6. Munsami A, Gouse H, Nightingale S, Joska JA. HIV-Associated Neurocognitive Impairment Knowledge and Current Practices: A Survey of Frontline Healthcare Workers in South Africa. Journal of Community Health. 2020 Jul 29:1-7.

7. Liboro RM, Ibañez-Carrasco F, Rourke SB, Eaton A, Medina C, Pugh D, Rae A, Ross LE, Shuper PA. Barriers to addressing HIV-associated neurocognitive disorder (hand): community-based service provider perspectives. Journal of HIV/AIDS & Social Services. 2018 Jul 3;17(3):209-23.

6. PLOS authors have the option to publish the peer review history of their article (what does this mean?). If published, this will include your full peer review and any attached files.

Reviewer #1: No

Reviewer #2: No

---

## [Author Response · Author response to Decision Letter 0]

4 May 2021

Réponses to academic editor and reviewers:

Dr Reppermund additional requirements:

- Apologies, I have gone through the manuscript and changed the labelling for my supporting information (S1 Table) and Fig 1 label. I have also changed headings to reflect the Heading 1, 2 etc. requirements.

2. Thank you for stating the following in the Disclosures section of your manuscript:

"J H V reports honoraria and research grants in trials sponsored by Merck, Janssen Cilag, Piramal and Gilead Sciences. SB reports grants and personal fees from Abbvie, personal fees and non-financial support from Lilly, personal fees from Eleusis, personal fees from Daval International, personal fees from Boehringer-Ingelheim, personal fees from Axovant, personal fees from Lundbeck, personal feesfrom Nutricia, outside the submitted work; he has been employed by the Department of Health for England. These funders had no role in the design of the study; in the collection, analyses, or interpretation of data; in the writing of the manuscript, or in the decision to publish the results."

Please remove any funding-related text from the manuscript and let us know how you would like to update your Funding Statement.

- I have removed all funding related text from the manuscript i.e. funding statement and disclosures statement). I can confirm this does not alter our adherence to PLOS ONE policies on sharing data and materials. In addition, we do not have any competing interests associated with this work.

3. Please include captions for your Supporting Information files at the end of your manuscript, and update any in-text citations to match accordingly.

- This has been corrected.

Reviewer 1 comments:

1. The authors use the phrases “PLWH with HAND” and “PLWH and HAND” interchangeably. Either or are fine, but can the authors please be consistent? 

- This has been changed to read ‘PLWH with HAND’ throughout

2. There are some minor grammatical errors throughout the manuscript

- Apologies, errors have been changed and the manuscript checked thoroughly.

3. The sub-heading “search strategy” on line 174 should be changed to “search results” or “search strategy outcomes” since the actual results of the search strategy are discussed in this section. 

- This has been changed to ‘Search results’

4. The authors use the PRISMA diagram to visually represent their results and also provide the PRISMA checklist for scoping review, but have chosen a framework by Arksey and O’Malley (2005) to guide their methodology. Please can the authors elaborate on this choice and explain why they deem it appropriate to apply this framework in conjunction to the PRISMA statement.

- The PRISMA statement for scoping reviews was developed in part based on the framework developed by Arskey & O’Malley. The PRISMA guidelines provide a more detailed outline of the process first reported by Arksey & O’Malley. As such we have followed the framework by Arksey &O’Malley, but sought further guidance from the PRISMA extension for scoping reviews. I have included a sentence on line 130-132 to clarify this.

5. Please can the authors include in the manuscript, a statement on what makes a scoping review different from a systematic review when the ultimate goal of both is to describe the depth and breadth of the state of science in a particular area/topic and how this review fits into that definition.

- We have included on line 115 ‘A scoping review was selected based on preliminary searches showing that studies examining QoL in PLWH with HAND were lacking, with the majority of studies found either not examining QoL (or any dimension associated with QoL) or not look at those PLWH with HAND. Munn et al (14) states the utility of scoping reviews, as opposed to systematic reviews, when examining limited, disparate, or emerging evidence, and broad as opposed to specific questions are posed. These indications allow for questions regarding types of available evidence, how the research has been conducted, key factors of characteristics of a topic, and the identification of gaps in knowledge, and as such, considered more appropriate for this research topic’ 

6. In the PRISMA diagram, please check for a typo in the block “records after duplicates removed n=460”. This number should be 547 as these are the records AFTER duplicates were removed. Alternatively change the text in the block to read “duplicate records removed n =460”

- Thank you, this has been changed to read ‘n = 547’.

7. The image quality of the PRISMA diagram is not very good, please can the authors provide a 600dpi image or use the word template for PRISMA diagrams provided on the PRISMA statement website

- We have improved the image quality to 600dpi

8. Results. Table 2 of the results provides a very good overview of the studies included and their findings. The authors report that 4 of the included studies did not include gender distinctions. Given that some neuropsychological tests take gender into consideration when doing the scaled scoring it would be important to check which neuropsychological tests these studies used to classify HAND. Regardless of the study aims, standard neuropsychological test scoring according to the test manuals needs to be observed, so it may be that these studies did not report the gender distinctions as opposed to it being a methodological matter. 

- This is a great point, we have included the following statement on line 295 ‘None of the studies report whether neuropsychological testing considered gender distinctions for normative scores or whether it was considered in diagnosing HAND, however this would normally only be examined in association analyses which seek to examine gender distinctions in HAND.’

9. For the Canadian study that was included regardless of the treatment status of the participants, it is confirmed that the data reported in that study was collected in the accessible ART era even though the publication date was 2018?

- Yes, we have contacted the study team and they confirmed interviews took place in 2015 and all participants had access to cART. This has been amended in the manuscript on line 307 to read “One study was included in this review(48), despite not providing ARV status information (likely due to its qualitative methodology and that it was not deemed relevant for the objectives of the research) however, contact was made with the study authors who confirmed all participants had access to cART”.

10. What the discussion is missing is the closing of the circle for the case to include quality of life into the 90-90-90 WHO targets. The authors identify the main strength of this review as being the systematic way in which it was conducted. While I agree that this is a strength of this review, I feel that this secondary to the very important dialogue this review has paved way for. The results, and the argument created in the discussion make a very strong case indeed for having quality of life added into the 90-90-90 targets and this should be emphasized more in the discussion.

- Thank you very much for this point – we have included the following in our conclusion. I hope closes the circle and emphasizes the necessity of the fourth 90.

‘QoL outcomes have received increased focus over the last two decades, and recent calls to include QoL in The WHO’s HIV/AIDS 90-90-90 target is justified based on multiple sources of evidence finding significant impacts of living with HIV on QoL(73-75). While there is more need for research exploring QoL in those living with HAND, the emerging evidence detailed in this review along with evidence from those studies reporting poorer QoL in PLWH broadly, shows that a paradigm shift is required when thinking about those living with HIV. Clinical care needs to reflect the realities of patients’ lives and multidimensional support is required to ensure treatment effects are maximized and reflective of factors most important to patients.’

11. I do not agree that the stringent criteria applied to the assessment of cognitive impairment is a limitation of this study. This should be added to the strengths of the study because these criteria ensure that a proper gold standard method was used to assess HIV-associated cognitive disorders in the included studies. 

- This has been changed on line 557 to read ‘Furthermore, we applied stringent eligibility criteria to the assessment of cognitive impairment in studies included, ensuring that the research ‘gold-standard’ Frascati Criteria (6) was used to assess HAND’.

12. A possible limitation of this study is the paucity of studies from sub-Saharan Africa. The impact of socioeconomic status on quality of life has not been acknowledged, and while other reviews have been excluded from the search strategy, the authors should acknowledge a recent review and meta-analysis published in 2020. Although this review does not specifically assess quality of life in HIV+ persons with cognitive impairment, important assertions are made regarding social and other determinant of quality of life in PLWH.

Ghiasvand, Hesam, Peter Higgs, Mehdi Noroozi, Gholamreza Ghaedamini Harouni, Morteza Hemmat, Elahe Ahounbar, Javad Haroni, Seyran Naghdi, Ali Nazeri Astaneh, and Bahram Armoon. "Social and demographical determinants of quality of life in people who live with HIV/AIDS infection: evidence from a meta-analysis." Biodemography and social biology 65, no. 1 (2020): 57-72.

- We agree, thank you for raising this important point. This has been changed on line 571 to read ‘Limitations of this review include the paucity of studies from sub-Saharan Africa, where the burden of HIV is highest, along with a lack of examination into the sociodemographic factors impacting QoL in this population.’

- And we have acknowledged the Hesam et al (2020) study by including (on line 534): ‘Interestingly, a recent meta-analysis examined the sociodemographic impacts on QoL in PLWH: finding low socioeconomic status, stigma, less education, and younger age to negatively influence QoL, and increased social support to positively influence QoL(72). This study emphasizes the importance of understanding sociodemographic influences on QoL in PLWH with HAND particularly given the intersectional and compounding issues facing this group’. 

Reviewer 2 comments:

1. First, HIV Associated Neurocognitive Disorder (HAND) is not a clinical diagnosis. It is used for research purposes. The authors should make this important distinction. ICD-10 or DSM-V clinical diagnoses for HIV-associated neurocognitive problems require different criteria than the Frascati criteria

- Thank you for raising this important distinction. We have changed some of the wording in the introduction and reported distinctions between methods (line 61-72) to acknowledge this, the section now reads ‘‘HIV-associated Neurocognitive Disorder (HAND) is the term coined to refer to clinically significant declines in multiple domains of neuropsychological functioning that are not exclusively attributed to factors other than HIV infections. For research purposes, the gold-standard Frascati criteria (6) is commonly used to identify HAND across three thresholds of impairment (asymptomatic Neurocognitive Impairment (ANI); mild neurocognitive Disorder (MND); and HIV-associated Dementia (HAD)). Frascati Criteria-based HAND involves testing across broad cognitive domains, with HAND identified when two or more domains are one or more standard deviations below normative scores and for MND and HAD when daily functioning is impacted. Notably, for a clinical diagnosis of HAND (based on Diagnostic and Statistical Manual of Mental Disorders (5th ed; DSM-V), impairment but be reported both objectively (e.g., decline in standard neuropsychological testing) and subjectively (e.g., complaints), whereas evidence of subjective impairment is not required for Frascati-based HAND identification. 

2. Second, the authors may wish to consider discussing that screening and diagnosing HIV associated NCIs remain very limited in many clinical settings as do services for HAND

- We agree this is an important consideration and have included the following sentences on line 573 “Furthermore, important consideration must be paid to the context in which this research appears, whereby screening and diagnosing HIV-associated neurocognitive impairments remains relatively limited in clinical settings(2, 73), with no consensus on screening tools and concerns regarding the importance of routine testing(74, 75), given the majority of instances are asymptomatic and services extremelylimited(76).’

Reference list changes:

In response to the reviewers comments the following references have been included: 

Tricco AC, Lillie E, Zarin W, O'Brien KK, Colquhoun H, Levac D, et al. PRISMA Extension for Scoping Reviews (PRISMA-ScR): Checklist and Explanation. Ann Intern Med. 2018;169(7):467-73.

- This was included in response to Reviewer 1, point 4

Munn Z, Peters MDJ, Stern C, Tufanaru C, McArthur A, Aromataris E. Systematic review or scoping review? Guidance for authors when choosing between a systematic or scoping review approach. BMC Med Res Methodol. 2018;18(1):143.

- This was included in response to Reviewer 1, point 4.

Ghiasvand, Hesam, Peter Higgs, Mehdi Noroozi, Gholamreza Ghaedamini Harouni, Morteza Hemmat, Elahe Ahounbar, Javad Haroni, Seyran Naghdi, Ali Nazeri Astaneh, and Bahram Armoon. "Social and demographical determinants of quality of life in people who live with HIV/AIDS infection: evidence from a meta-analysis." Biodemography and social biology 65, no. 1 (2020): 57-72.

- This was included in response to Reviewer 1, point 12.

Shey ND, Dzemo KO, Siysi VV, Ekobo AS, Jelil NA. Quality of life of HIV patients on highly active antiretroviral therapy: A scoping review. Journal of Public Health and Epidemiology. 2020;12(1):63-73.

Miners A, Phillips A, Kreif N, Rodger A, Speakman A, Fisher M, et al. Health-related quality-of-life of people with HIV in the era of combination antiretroviral treatment: a cross-sectional comparison with the general population. Lancet HIV. 2014;1(1):e32-40.

Engelhard EAN, Smit C, van Dijk PR, Kuijper TM, Wermeling PR, Weel AE, et al. Health-related quality of life of people with HIV: an assessment of patient related factors and comparison with other chronic diseases. AIDS. 2018;32(1):103-12

- These were included in response to Reviewer 1, point 10.

American Psychiatrict Association. Diagnostic and statistical manual of mental disorders. 5th ed. Washington, DC, 2013.

- Include in response to Reviewer 2, point 1. 

Haddow LJ, Accoroni A, Cartledge JD, Manji H, Benn P, Gilson RJ. Routine detection and management of neurocognitive impairment in HIV-positive patients in a UK centre. Int J STD AIDS. 2013;24(3):217-9.

The Mind Exchange Working Group. Assessment, diagnosis, and treatment of HIV-associated neurocognitive disorder: a consensus report of the mind exchange program. Clin Infect Dis. 2013;56(7):1004-17.

Morley D, McNamara P, Kennelly S, McMahon G, Bergin C. Limitations to the identification of HIV-associated neurocognitive disorders in clinical practice. HIV Med. 2013;14(8):497-502

- These were included in response to Reviewer 2, point 2

---

## [Editor Report · Decision Letter 1]

6 May 2021

Quality of life in people living with HIV-associated neurocognitive disorder: A scoping review

PONE-D-21-03610R1

Dear Dr. Alford,

We’re pleased to inform you that your manuscript has been judged scientifically suitable for publication and will be formally accepted for publication once it meets all outstanding technical requirements.

Kind regards,

Simone Reppermund, PhD

Academic Editor

PLOS ONE
---

## [Editor Report · Acceptance letter]

10 May 2021

PONE-D-21-03610R1 

Quality of life in people living with HIV-associated neurocognitive disorder: A scoping review study 

Dear Dr. Alford:

I'm pleased to inform you that your manuscript has been deemed suitable for publication in PLOS ONE. Congratulations! Your manuscript is now with our production department. 

Kind regards, 

on behalf of

Dr. Simone Reppermund 

Academic Editor

PLOS ONE